# The Role of Cultural Heritage in Promoting Urban Sustainability: A Brief Review

Sanober Naheed [1] and Salman Shooshtarian [2,*]

1 Department of Geography, Adi Keih College of Arts and Social Sciences, Adi Keih, Eritrea
2 School of Property, Construction and Project Management, RMIT University, Melbourne 3000, Australia
* Correspondence: salman.shooshtarian@rmit.edu

**Abstract:** Cities are hubs of social and cultural activity, and culture is key to what makes cities creative, and sustainable. The Post-2015 Development Agenda has prioritized culture and how it may help people and communities create the future they desire. The study aims to determine the link between cultural heritage and urban sustainability and how multidisciplinary education can help organize urban issues. The article is of relevance to the emerging multicultural urban society with wide socio-economic disparities straining the global urban resilience and posing a challenge for future policy implementation. A systematic literature review was conducted using the Dimension database, and the results were analyzed using VOS viewer. The study also employed the PRISMA quantitative approach for selection criteria. This paper has identified understudied themes including community heritage, sustainable urban governance, and behavioral and multidisciplinary approaches. It is strongly felt that undertaking this study will not only add to the literature in cultural heritage study but also help further multidisciplinary and knowledge-based inquiry, which is currently evolving in the academic domain. Therefore, urban academics have a duty to resolve the issue confronting global urban sustainability and cultural disputes. Future research is required to simplify the current complex issue to make it more relevant and inclusive.

**Keywords:** urban sustainability; cultural heritage; multicultural; community cohesion; knowledge-based urban development; multidisciplinary

## 1. Introduction

Cultural heritage, both tangible and intangible, has come to play an essential role in developing strategies for enhancing the image of contemporary cities [1]. The Dialogues on the Post-2015 Development Agenda have given a central place to culture and contributed to people and communities creating the future they would prefer [2]. With the adoption of the 2030 Agenda, the world community acknowledged—for the first time—the role of culture in long-term development. The 2030 Agenda implicitly refers to culture across many of its Goals and Targets [3]. Global urbanization is directly correlated with an increased understanding of typical urban challenges [4], fostering creativity, wealth creation, social expansion, and the use of human and technology resources, culminating in unprecedented economic and social advancements [3,5]. Over time they have evolved, giving each location a distinct identity and heritage. At the same time, cities struggle to restructure whilst retaining their distinct identity and historical connections [6,7]. The dialogues emphasize heritage as a vital tool for dealing with current urban realities [2]. They provide examples of past urban landscapes as the key to creating cities that are sustainable, livable, and inclusive [8]. The task of integrating heritage and ensuring its significance in the context of sustainable urbanization is to demonstrate that heritage contributes to social cohesion, well-being, creativity, and economic appeal, and is a factor in improving community understanding and benefitting future generations [8,9].

Cultural heritage plays a vital role in formulating action plans (Figure 1) for the modern city and increasing urbanization [10]. The publication of the New Urban Agenda

is another notable document that has added more impetus to urban sustainability and culture [5,8] and the inclusion of urban sustainability in policy frameworks such as the Sustainable Development Goals (SDGs). SDG No. 11 explicitly calls for "making cities and human settlements inclusive, safe, resilient and sustainable" [5,11]. Scholars [12] have also focused on "Spaces of Possibility" (SoPs) and linked Institutional Innovations (II) as tools to investigate how cultural actors and urban policymakers might propose new approaches that emphasize the cultural component of sustainability through a selection of recent cases of urban cultural practices.

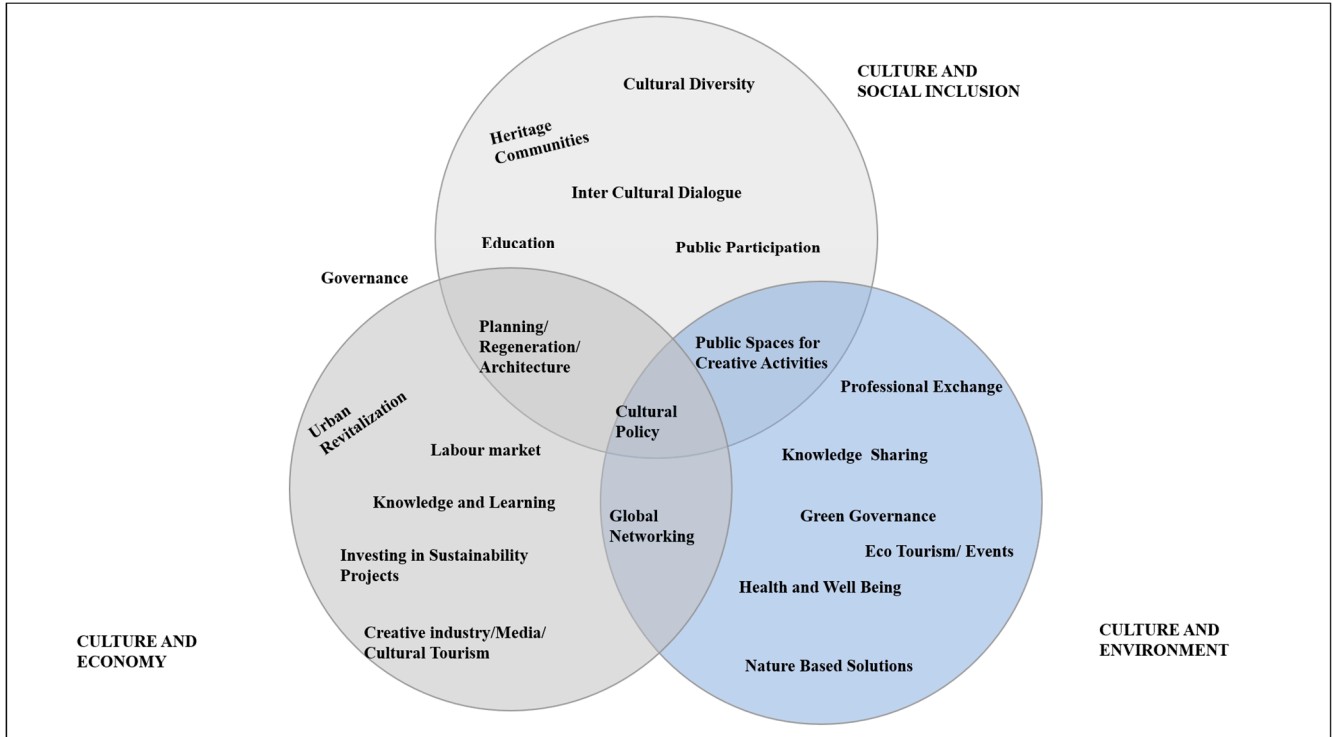

**Figure 1.** Culture and Sustainable Development Action Plan. Source: Author.

Author [13] claims that the inefficiency with which cultural heritage is managed should be blamed on government failures to address public policies for the sustainability of cultural heritage or cultural policy independently, which are often included in the co-ordinated governance of cultural heritage. Marked inequalities and injustices emerged [14] within urban societies during the COVID-19 crisis that impacted the labor market and education progress. The world has witnessed protests against injustices and government failures in addressing citizens' basic rights. In this paper, an attempt has been made to investigate how culture is related to the development of sustainable communities and neighborhoods, as well as a multidisciplinary and knowledge-based inquiry into the cultural aspects of sustainable urban management. It is also vital to discuss the challenges that come with governing the urban system. It is strongly felt that undertaking this study will not only add to the literature on cultural heritage study but will also help develop more clarity of urban sustainability approaches, which are currently evolving in the academic domain.

The review thus far has helped identify a scholarly gap in the urban sustainability literature with regard to culture [12,15]. Despite the fact that research on cultural heritage and sustainability has grown in popularity since 2008 [1], only after 2015 did a well-developed method emerge [6]. This review focuses on the very basic relationship between culture and sustainability and discusses how culture contributes to the development of the urban landscape.

Objectives and Structure

This review study attempts to achieve the following objectives by conducting a systematic literature review of the existing literature:

1.  To understand how the social and cultural elements of urban life help to address the future challenges of towns and cities.
2.  To explore how local cultures can contribute to the sustainable development of cities.
3.  To examine how interdisciplinary education might help organize urban issues.

This paper briefly conceptualizes the two aspects of cultural heritage and urban sustainability, used in this work followed by the methodology. It is hoped that this work will inspire future academics to focus on the growing cultural diversity in the global context, as well as seeking solutions through multistakeholder and multidisciplinary networks. Finally, it discusses how cities can maintain their unique culture and the current state of knowledge and practice for future sustainable cities.

### 1.1. Culture in the Global Agenda for Sustainable Development of Cities

Cultural legacy has gained international attention, principally owing to the efforts of the United Nations Educational, Scientific, and Cultural Organization (UNESCO) [1]. Inclusion of culture as a priority component of urban plans and strategies is also referred to in Transforming our World: The New Urban Agenda. A growing awareness related to cultural factors in city prosperity and its adoption by United Cities and Local Governments (UCLG), is intended to bridge the gap between urban cultural policy and sustainable development [16,17]. Culture and cities are so inextricably interwoven that sustainable cities provide access to culture in the 2030 Agenda for Sustainable Development [3].

Goal 11 of the Sustainable Development Goals has a specific aim of "intensifying efforts to maintain and safeguard the world's cultural and natural heritage" [3,5]. In addition to SDG 11, which focuses on urban spaces and functions, all 17 SDGs are important to cities.

Academics use heritage, innovation, and intercultural dialogue to inspire new planning and governance models to inspire the creation of new jobs, address social inequities, minimize urban disputes, and lower cities' ecological imprints, making cities more inclusive and resilient [3,5,17,18]. UNESCO is building on the various components of culture—tangible and intangible heritage, the creative economy, cultural tourism, museums, and other local cultural organizations—by focusing on the distinct components of culture. A broader holistic view places culture as a social, cultural, and economic resource for long term city development [3].

The significance of cultural heritage and the utilization of cultural assets in the promotion of long-term urban development is advocated by a number of scholars [6,19,20]. Some scholars consider cultural heritage to be a tool for cultural legacy, creativity, and diversity that can effectively contribute to ethical, inclusive, integrative, and extended urban growth [6]. Cities have also been identified as laboratories [11,21,22] that are best equipped to appreciate cultural diversity to promote community resilience and well-being, while also promoting environmental sustainability. The dynamics of tangible and intangible heritage are considered a driver and enabler of economic, social, and environmental benefits and sustainability [20]. As a driver, culture directly contributes to economic growth and societal benefits. As an enabler, it improves the efficacy of growth initiatives [23].

Heritage sustainability thus can be achieved through innovative solutions and civic participation, since heritage is part of citizens' daily lives [13]. An effective measure in this regard is that the international community has recognized culture as a fundamental component of effective urban planning, and a vital breakthrough for the definition of a New Urban Agenda [8].

### 1.2. Urban Sustainability

The growing emphasis on cities and their role in achieving sustainability is expected since cities currently host more than half of the world's population [5]. Academics have therefore attempted to establish indicators, evaluation techniques, assessment tools, and rating systems for integrating sustainability in urban planning and development [24]. The

author of [5] clearly states the multiple benefits attached to assessment, such as allowing planners and policymakers to measure targets, promoting openness and accountability, boosting awareness, expediting development proposal and approval procedures, and allowing better-informed decision making. Some have used the geo-design paradigm to promote the understanding of urban components that impact vitality and urbanity [25]. Additionally, the diverse dimensions of urban sustainability are apparent through studies on biophilic design and planning [26] which have the potential to provide an indicator to an urban environment's sustainability implications. Others have used the Historic Urban Landscape (HUL) approach [7] to determine the sustainability of urban design in a post-disaster situation.

A more precise definition of sustainability considers the interplay of environmental, social, cultural, and economic factors (Figure 2), designed to promote the balanced development of the environment, society, and economy [27,28] and safeguard fundamental public liberties [29]. Each factor is related and urban sustainability can be realized by the complex interaction of these three scopes [8]. Environmental sustainability entails safeguarding natural resources and ecosystems [29]. It is also referred to as ecological sustainability. Social sustainability seeks to maintain and enhance one's social standing (wellbeing and health), as well as maintain the balance and cultural diversity of social systems. Cultural sustainability is associated with the preservation and development of cultural values such as the cultural heritage, cultural life, and cultural activities that should be passed down to future generations. Economic sustainability refers to achieving the optimum flow of economic well-being while protecting current savings [30].

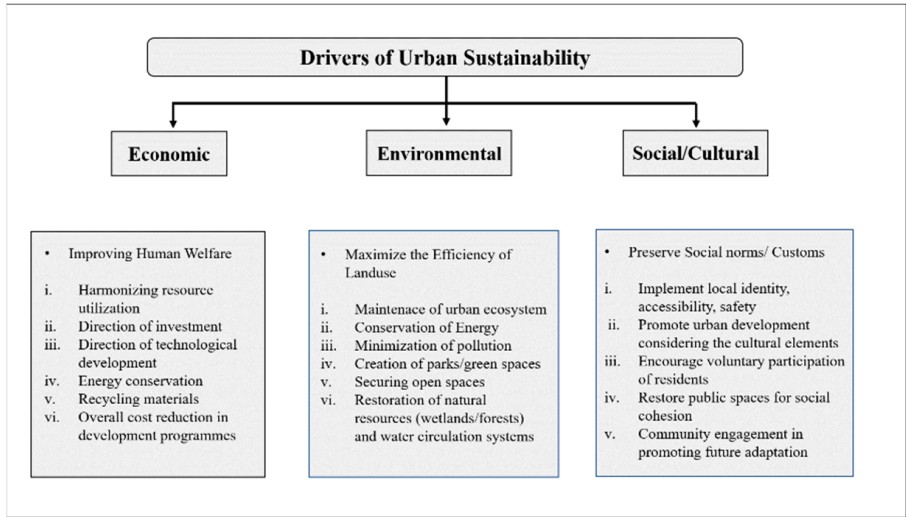

**Figure 2.** The current society's sustainable development concept is explained as covering economic, environmental, and socio-cultural factors. Source: Author.

Therefore, no single definition of a sustainable city exists [31]. As cities continue to grow and new ones evolve, sustainability can be achieved through institutional directives and economic revival. All parties, including central and local governments, non-governmental organizations, and civil societies, have a responsibility towards making urban landscapes more sustainable and livable, as well as fostering a sense of belonging by harboring these environments with a distinct, unique identity [30].

An understanding of urban sustainability through culture is made possible via tangible and intangible heritage, creative enterprises, and other forms of artistic expression in fostering inclusive social and economic growth [32]. The economic sustainability of cultural heritage is often assessed through the development and management of sustainable tourism relevant to local communities. Social impacts are not only an influence but are rather a community-defined value system. Environmental sustainability seeks to emphasize the impact of climate change on heritage and the adoption of circular and green economic

principles [13]. Culture may also address both social and economic issues in its role in poverty alleviation [6,33].

*1.3. Challenges to Urban Sustainability*

Owing to the topic's tremendous complexity, there are ambiguities in its definition, execution, and quantification [34] considering that urban sustainability is a place-specific concept [31]. Cities have developed into transformative platforms, the full potential for which still remains untapped. The objective of a new people-centered city through culture-sensitive urban policies is supported by access, representation, and participation principles and inclusive processes [6]. Interest in urban revitalization [21] is focused on historic city centers [35], vital to accomplishing long-term urban development. This is largely evolving from the idea of compact cities and changing land use patterns [36]. Studies have used decision support tools such as the Urban Transformation Matrix [35] and SWOT (Strength, Weakness, Opportunities and Threats) [30], along with circles of sustainability for heritage preservation [21]. The authors claim architectural heritage is best preserved when social issues are considered [7,30].

In this context, city resilience is particularly relevant in regard to managing the conflict between cultural heritage preservation and urban regeneration. Academics have maintained the necessity to study and comprehend urban resilience, citing a new global context marked by dynamic urbanism [6]. The link between intangible cultural heritage and urban resilience discourses is relatively recent [1], with academic research on the subject constantly expanding since 2017 [1,15,18].

Conflicting ideas also prevail in regard to urban governance. Academics [34] have noted that there is no thorough systematic understanding of the interaction between government and citizens for sustainable development in its social, economic, and environmental dimensions, as outlined by the United Nations 2030 Agenda. The challenge to poorly-equipped urban governing systems emerge in the face of emergencies [37] such as pandemics. Scholars and practitioners have frequently emphasized the relevance of involvement in policy-making and implementation in the context of sustainability governance. This is thought to promote mutual learning, bringing together different parties in the formulation of policy ideas and providing more successful outcomes [34]. The authors of [38] have also attempted to explore distinct instruments for participatory approaches in the framework of cultural heritage governance. Using the notion of managing through networks and cohesiveness as a starting point. The authors apply this method to cultural heritage by examining how heritage is defined in different governance frameworks and what types of responsibilities individuals have in different forms of heritage governance. They find the area highly theoretically lacking in practice and recommend a co-creative approach that strives to encourage the local community to participate in heritage processes, and that necessitates collaboration among professionals, managers, stakeholders, and members of the communities affected by heritage regulations. The co-creative method provides the transparency to identify changes early and establish adaptable decision-making procedures due to collaborative activity and leadership initiatives in the public domain [38]. Participatory governance necessitates community efforts, but they can only succeed if the local government is actively involved in facilitating partnership formation and enforcing the rules, therefore enhancing the legitimacy of activities [38]. A limitation to this initiative is visible in the lack of allocation of funds and co-ordination between the different tiers and scales of governance, especially in developing countries [37].

The reality of failed co-ordination surfaced during the COVID-19 pandemic and presented a severe threat to long-term sustainable development [16–39]. A significant impact has been felt on culture, education, and tourism; with schools being shut in 180 nations, more than three-quarters of the world's 1.5 billion pupils have been prevented from attending [39]. The poorest and least-educated have been impacted the most and pose a major challenge to SDG1, the objective of eradicating poverty by 2030 [5,39]. Studies [5,39,40] emphasize that future research and local government policies should consider the negative consequences

of lockdowns on loneliness, mental health, senior mobility, violence, and other factors to estimate the impact of the pandemic on cities [40]. The pandemic is projected to radically impact future city management and governance. Actions taken in the years ahead will decide whether post-COVID-19 cities are designed and governed in a more sustainable manner [40]. Marked progress in the urban future can be directed to urban academics for actively participating in constructing the "global urban imagination" that supports global urban governance [37].

## 2. Methodology

For the purpose of this review, the Dimension database was used to collect relevant studies outlining urban sustainability and cultural heritage relationships. Other search engines were not used, due to a lack of access. The library within the authors' reach has access to JSTOR, so a further search was conducted on JSTOR using the search terms "Sustainable City" or "Cultural Heritage". A mixed result was obtained and manual sorting was performed to retrieve the most relevant work, eliminating articles from medicine and agriculture. Of the 321 results obtained from Dimension, 162 were selected, which on further screening of abstracts were reduced to 52 peer reviewed articles. A systematic literature review (SLR) was carried out using the Dimension database with the goal of achieving high-quality results whilst ensuring objectivity. The data were further analyzed on VOS viewer software to map the results on co-authorship and keywords and countries of publication. Since all the required literature was not available, JSTOR and Google Scholar were also used to retrieve the most relevant literature to the study structured on culture and urban sustainability. This brief literature review was improved by the examination of similar approaches.

Table 1 summarizes and quantifies the data acquired using the given search strategy, giving a summary of the major findings in each phase that led to the final sample. Further analysis of the gathered results was performed. This method involves bibliometric analysis, which provides for a quantitative perspective, as well as a qualitative analysis using the VOS Viewer software (version1.6.18) followed by a content analysis.

**Table 1.** Quantification of data.

| Inclusion and Exclusion Criteria | Number of Results |
|---|---|
| Availability | 32 |
| Suitability of results to research question | 162 |
| Inclusion of peer-reviewed results | 52 |

*Data Screening and Analysis*

To manage data collection and screening, the study used a PRISMA [41] (Preferred Reporting Items Systematic Reviews and Meta-Analyses) diagram. The diagram, as shown in Figure 3, consists of four major steps. 1. Identification: approximately 321 articles dating between 2015 and 2022 were collected with some exceptions. The post-2015 dialogues on sustainable development were the main focus. Most were identified in the database and compiled from other sources as mentioned. 2. Screening: to remove duplicates and select only articles suited to the study and published in only peer-reviewed journals. Next, the abstracts and titles were scrutinized and 162 articles were separated for the review. 3. Eligibility: the evaluation criteria based on the selection criteria had the final number reduced to 60. 4: Inclusion: after reviewing the full text, 52 papers were finally included for review.

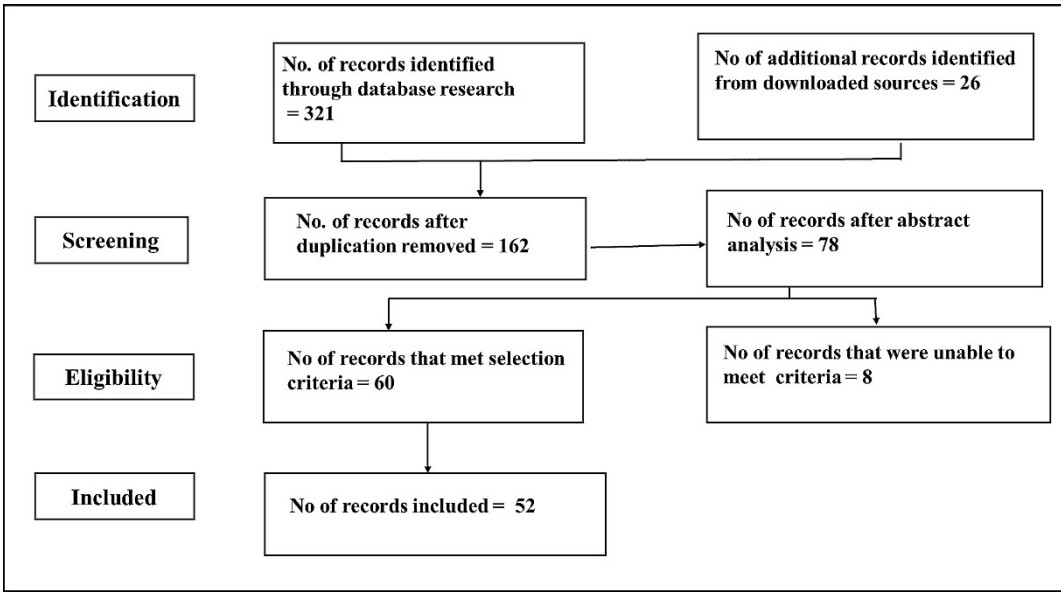

**Figure 3.** A flow diagram of PRISMA [41].

### 3. Results Analysis

*3.1. Bibliometric Analysis*

Quantitative data analysis permits an unbiased evaluation of the scientific characteristics of the publications' fixed parameters. To meet the research needs a bibliometric analysis of the 52 results was performed to ascertain the quantitative characterization and classification of the results. The analysis in VOS viewer helped derive the relationships between authors and co-authorship. Concerning the type of work, at the very outset peer-reviewed journal articles were selected. The journals having more than one publication for the related time period were few, with Sustainability having the most results (10) followed by Sustainable Cities and Societies (5) and The Science of the Total Environment, City Culture and Society and Journal of Cultural Heritage Management and Sustainable Development (2) each. Of the 21 results, Sustainability was the most prominent journal regarding the research topic.

According to the data collected, the majority of the authors appear to be linked with only one publication. The exception to this is Ayoob Sharifi [5,18] whose name appears to be associated with two publications in co-authorship with other authors. A co-authorship analysis of the authors obtained from articles with the greatest connections is shown in Figure 4. From the figure, the most prominent contributors to the field of urban sustainability have been obtained. Each colored cluster identifies with a prominent author and their contribution to the respective field of expertise and connection with others. However, they do not necessarily align with cultural heritage inquiry.

The term co-occurrence is a bibliographic analysis approach for detecting key emphasis areas and identifying terms of frequent occurrences clustered as conceptual groups. The size of each node corresponds to the number of times the word has occurred. The most immediate nodes are closely linked and the thickness of the links connecting them is proportional to the strength of the connection. Terms such as heritage, urban system, sustainable development goal, risk, urban planner, knowledge, urbanization, community, awareness, protection, transformation, and attitude had greater occurrence and total link-strength ratings, suggesting that they have gained more attention and are more closely related to the other terms. Figure 5 provides this analysis, for which 5 different groups were formed: green, red, blue, cyan and purple. On examining the terms occurring in each group, a thematic area can be identified connecting with heritage the central idea and the general topic of discussion. The topic concerns built heritage revitalization and urban planning (green cluster), assessment of the urban system related to sustainability development goals and research (blue cluster), community participation and awareness

through creative practices and transformative learning (cyan cluster), characteristics that emerge from the diversity of urban public space activities and have an impact on a people's sense of place (red cluster), and finally conservation and protection of heritage under the guardianship of UNESCO (purple cluster).

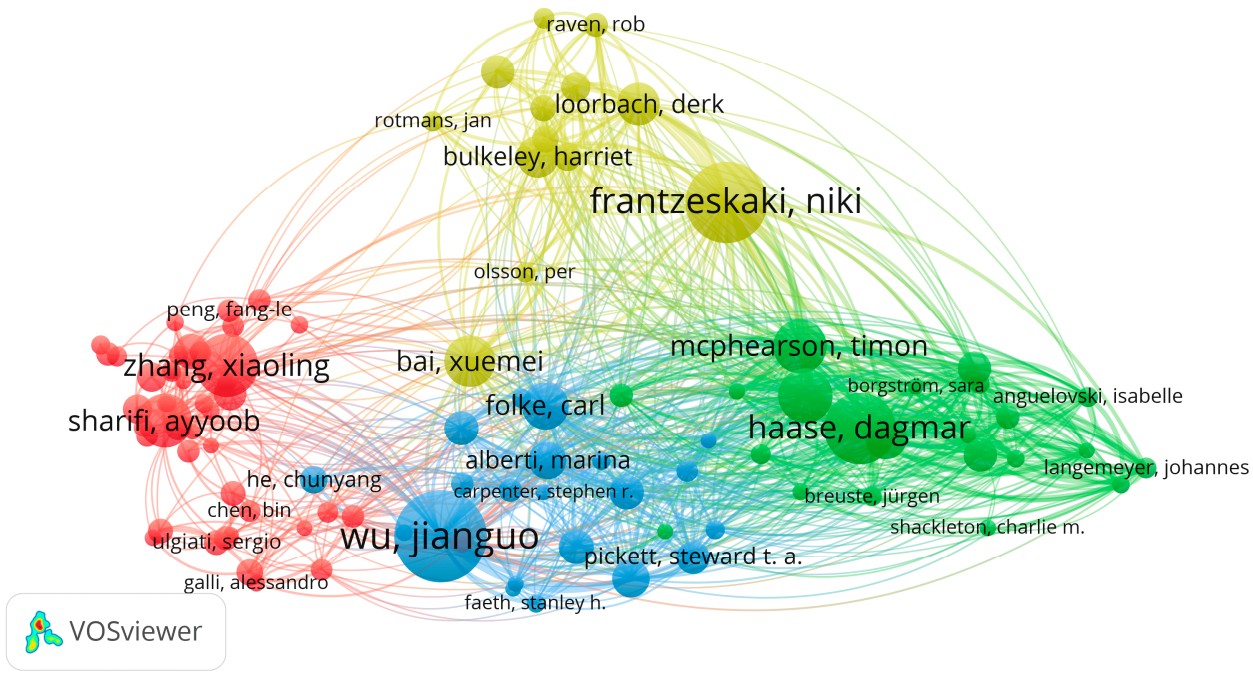

**Figure 4.** A co-authorship analysis (the node size is proportional to the number of citations).

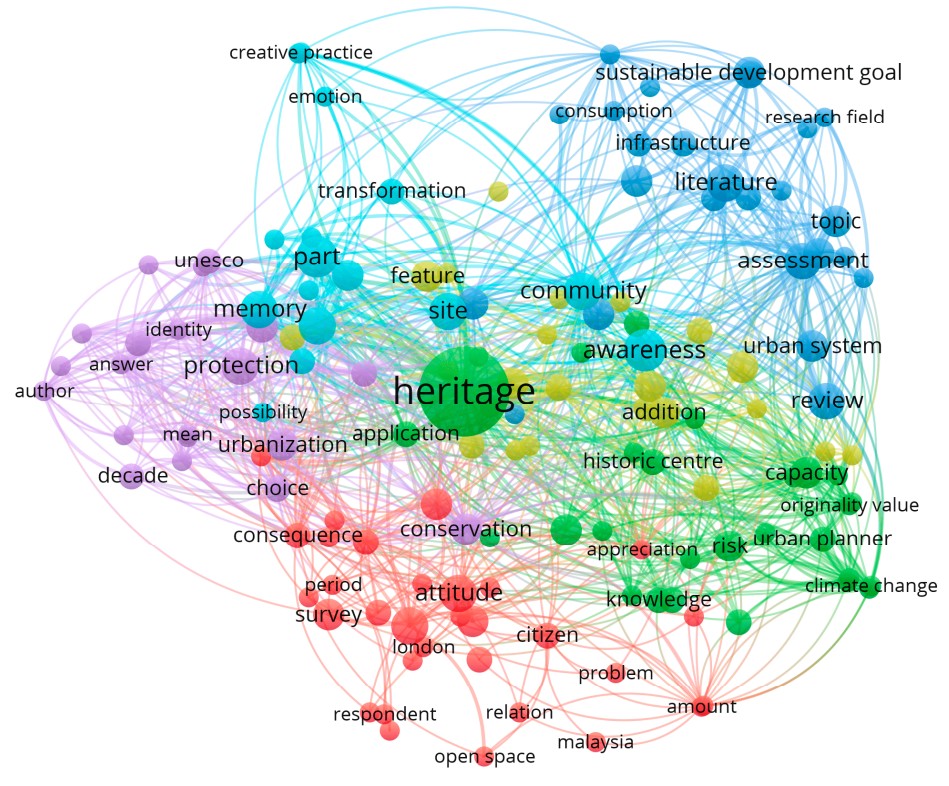

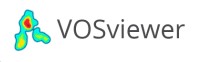

**Figure 5.** The term co-occurrence map (the node size is proportional to the number of terms).

Figure 6 shows which countries have contributed the most to the research field. The size of the node is proportional to the number of publications. China, the USA, the United Kingdom, the Netherlands, Spain, and Australia are dominant countries. While developed countries are major contributors, it is clear that some developing countries are also emerging. Given the rate of urbanization in developing countries, many countries from Africa and Latin America are missing. To properly inform the planning process, more research in the field is needed. With regard to the clusters, with occasional exceptions, strong connections between nations that are geographically nearby may be observed.

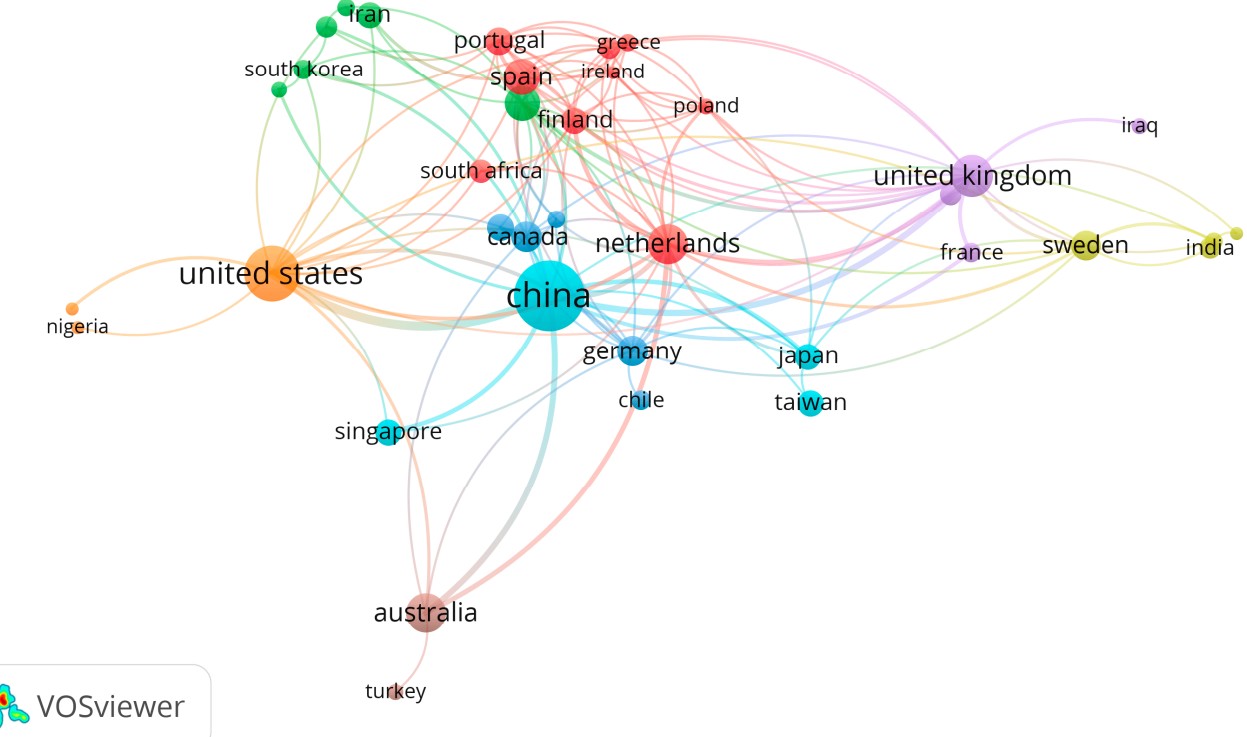

**Figure 6.** Country map (the node size is proportional to the number of publications).

### 3.2. Content Based Analysis

For this part, the collected data were read in their entirety to extract the key arguments offered by the publications relevant to cultural heritage and urban sustainability. After having read through the collected literature, three main areas of interest surfaced concerning urban heritage and sustainability studies: city uniqueness and cultural diversity, social connectivity and urban sustainability, and multidisciplinarity and city knowledge.

(a)   City uniqueness and cultural diversity

The socio-ecological response surrounding the cultural diversity within a city focuses mainly on historical, cultural, and social roots of the built environment. The concern has evolved as a result of the rapid pace of urbanization and global economic homogenization risking regional identities [6,30] and evolving geopolitical patterns [33] as a general perception.

The challenge also cuts across the transboundary migration of people with diverse ethnic origins. The advantages that it brings have been recognized [42] in re-building the urban infrastructure through the specialist contribution of professionals and culturally diverse individuals who serve as a valuable resource. Conversely, there are challenges involved, where xenophobia, racism, and exclusion of ethnic minorities are more apparent than ever [42].

In this context, the critical role of urban planners as facilitators of city resilience and the management of the conflict between cultural heritage preservation and urban regeneration

are apparent, citing a new global context marked by dynamic urbanism [1,15,29]. The competitive approach to urban planning, as emphasized in [43], should appeal to the creative class and provide a rich lifestyle, as well as the possibility to live as a part of a diverse community and culture.

Recognizing cultural heterogeneity as the driving element of new urbanism, a co-creative approach can help reduce segregation and encourage inclusion by popularizing each and every culture within the city's defining dimensions [6]. Some encouraging initiatives have come out of such attempts in establishing art museums such as the Centre Georges Pompidou in Paris and Massachusetts Museum of Contemporary Art. This has led other cities to follow and establish more specialized museums such as the Los Angeles Museum of Contemporary Art, the Museu d'Art Contemporani de Barcelona, and the Tate Modern in London. Similarly, the Reemdoogo Music Garden in Ouagadougou, Burkina Faso, has well-defined educational and cultural programs for youth. Such programs have the promise of promoting social cohesiveness within a culturally diverse urban setting [6], and also income generation and job-creation for some [42]. The creation of the Bilbao Guggenheim Museum presented the city with a unique opportunity to transform itself as a tourist center, sparking a massive economic boom in cultural tourism [42]. The primary objective is to inform policymakers to examine inclusive economic and social integration procedures so that culture can be properly recognized as a primary driving force for sustainable development [21,42,44].

Some instances can also be found outside of the West in the world's cultural, economic, and geopolitical systems that determine international cultural interactions and cultural policies. Sharjah was designated as the "World Book Capital" by UNESCO in 2019 [45] when this little Gulf city-state began to rebuild itself as an artistic and cultural center. Beijing's 798 art district has contributed to the transformation of the Chinese political capital's image into a vibrant art world. In 2017, the opening of Zeitz MOCAA museum in Cape Town confirmed its place as the hub of the African contemporary art scene. In 2018, the city of Buenos Aires partnered with the Art Basel Cities programme to fund a multiyear series of projects aimed at improving the city's art scene and connecting it with the global art world [45].

(b)　　Social connectivity and urban sustainability

Cultural heritage is an important aspect of community life and has an expression in social, economic, and environmental processes of cultural identity and religious adherences. Discourses on heterogeneous culture are characterized by the preservation of cultural legacy, in step with fixing memories in time so that the identity is preserved in the midst of contemporary globalized developments [8]. In the urban context, cultural legacy expresses and preserves the values and traditions of a city and its community, although its significance varies throughout communities and even among members of the same group [10,46]. It connects the past, present, and future while also posing the possibility of conflict [32].

In a city, diverse cultural groups may have distinct attitudes and perspectives on what is essential to their heritage, and hence assign different values to culture. The presence of these disparities can be a source of contention and, in some cases, can be the catalyst for activities that have a detrimental effect on cultural significance. Diversity is unavoidable, but it must be accepted and acknowledged to avoid potential disputes [8]. Understanding the cultural significance held by different groups within a society is an important step toward cultural tolerance, resolving problems through mutual understanding of value [46]. Some [25,47] attach the sense of place as a criterion to safeguard the identity of local communities and their intangible heritage. This eventually will help to maintain collective good and social cohesion, minimizing disparities at the same time. The role cultural legacies can serve is understood [3] to create social capital and social cohesiveness by providing a framework for involvement and engagement and aiding integration, boosting tourism, which supports overall economic well-being [47], a key feature of social inclusiveness [7]. In brief, cultural diversity can serve as an incentive for overall urban sustainability.

(c)     Local culture contributions to the sustainable development of cities

Although cultural heritage reliance on the local community has now recognized its proper implementation, in light of transformations as embedded in the 2030 agenda it is still inadequate [34]. The active role of local communities as representatives of tangible and intangible local heritage can follow on the lines of discussion present in the literature.

Evidence in support of local culture as a tool to achieve local sustainability goals [33,44] has also been acknowledged by the United Cities and Local Government (UCLG) network [47]. As part of culture, local wisdom and way of life contain many values of sustainability. It also improves the quality of life by encouraging a creative and productive economy [48] and necessitating societal commitment [49]. In this context, urban green and public spaces [50,51] can foster the growth of cultural activities, a sense of place and cohesion and cultural exchange [52,53]. Public spaces have always served as a conduit for the trade of goods, services, and ideas. They serve as venues for business transactions, multidisciplinary cultural expressions, social interaction, and peaceful coexistence. However, they are not yet well-developed in either theory or practice [10].

The transformative potential of sustainable urban behaviors that can strengthen both people's well-being and the psychological basis of sustainable cultures has remained understudied. Cultural values clearly define urban behavior, visibly so in transportation and mobility, as well as environmental use [2,6,54]. Researchers [54] find it difficult to develop a co-evolution between PEH (Pro Environmental Habits) and environmentally conscious identities, and propose social–ecological systems research, particularly among urban dwellers who are frequently psychologically alienated from the Biosphere.

Therefore, urban sustainability can be clearly connected with urban planning and related cultural policies. Culture is progressively being mainstreamed in a variety of policy sectors, including innovation, economic development, social stability, urban planning, and city-wide international relationships and policies [45,47]. The impact that cultural investment and creative professionals have on people's emotions and morale is mostly unknown because such activities are rarely identified as cultural [45,55]. Additionally, actionable plans at multiple scales and the adoption of a multidisciplinary approach including specialists from diverse sectors will be beneficial.

(d)     Multidisciplinarity and city knowledge

How might interdisciplinary education help organize urban issues?

To answer the above query, it is imperative for academics to create interest and follow it up in higher education, another focus area in the literature [56,57]. Cities have an impact on scientific urban research (Figure 7), contribute to urban sustainability, and shape cities [58]. Urban sustainability has evolved into a complex socio-cultural dialogue that advocates for higher education (HE). In this context, transdisciplinary techniques [58] are ideal for facilitating transformative learning in higher education [59], and integrating expertise and knowledge from diverse disciplines [56]. Two sets of inquiry can be developed, i. on the holistic approach and ii. on pedagogical challenges.

Wilkerson et al. [59] suggested a holistic approach to comprehend multidisciplinary endeavors and examine the need to enhance conceptual knowledge of the links between sustainable cities and development. This further demonstrates the significance of the multidimensional integration of social/cultural, economic, and environmental/climate variables in building sustainable urban models and policies.

Authors [56,57] emphasize exposing participants to a wide range of knowledge and world views, as well as a focus on reflexivity, which allows students to examine the roles of values, norms, and worldviews in defining, framing, and addressing sustainability challenges. Enriching urban sustainability studies through the use of multiple academic disciplines, as well as critical and analytical reasoning across subject areas [57], provides a solution for alternative and creative learning [56].

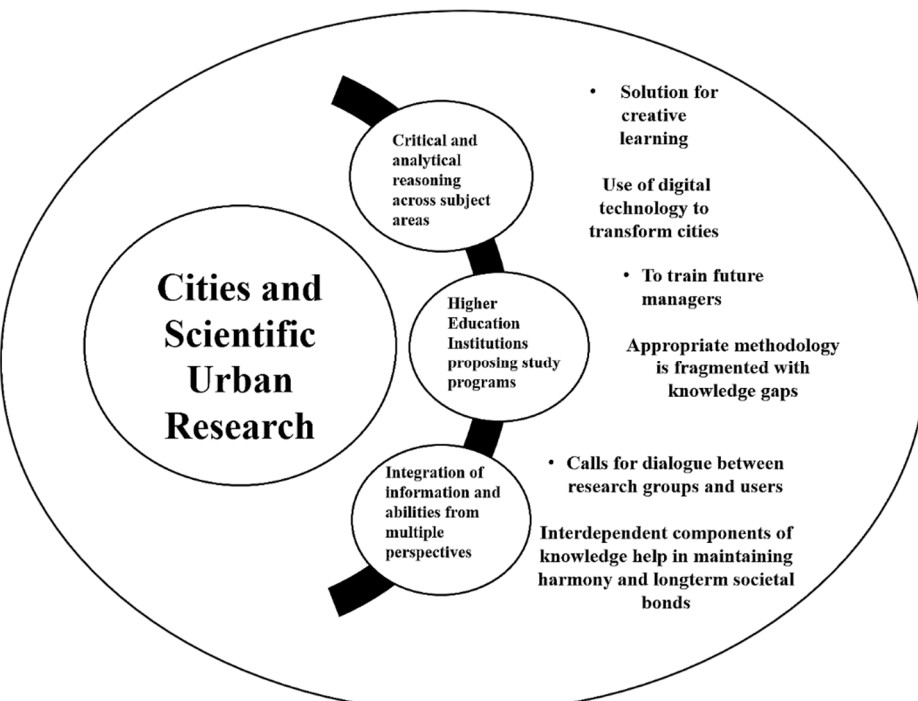

**Figure 7.** Prospects of a multidisciplinary approach to scientific urban research. Source: Author.

Academics have also shown concern over the existing knowledge gap in practical frameworks to assess methods for urban sustainability [60] that remains fragmented and lacks interdisciplinary perspectives [61]. Due to the growing relevance of such education, a number of university curricula have started to focus on establishing urban sustainability education programs and courses. Additionally, many Higher Education Institutions (HEIs) are proposing study programs to provide a complete understanding of the technological environment to train future urban managers [55]. The limitation for such an endeavor lies in the absence of standards on how to establish the competencies that should be prioritized [57].

In addition, interdisciplinary studies have a pedagogical argument: fragmentation hinders learning, and evaluations of educational programs have frequently urged for better coherence and integration of diverse fields. The successful creation of sustainable solutions in urban settings necessitates the integration of information and abilities from multiple perspectives. Consequently, the capacity to think beyond disciplinary and institutional barriers is needed [62], as significant obstacles emerge when attempting to bridge disciplines and conduct interdisciplinary and transdisciplinary research [63].

Authors [59] contend that a change to multidisciplinary education may necessitate significant alterations on the part of both instructors and students, as well as HE institutions. Teachers must be able to encourage communication across multiple viewpoints, discourses, and approaches to solving sustainability challenges, which necessitates an open mindset and a willingness to learn while interacting with contrary ideas. Because no single discipline can cover all elements of Sustainable Urban Development (SUD), multidisciplinary and transdisciplinary techniques are more feasible and the results may be ready for the future [64].

A meaningful and interdisciplinary educational program in the urban realm is therefore vital [4,56]. It also necessitates great organizational skills and the engagement of key stakeholders and communities who are often excluded from the knowledge-generating process. Incorporating intercultural comparative learning [59] into urban sustainability programs will, therefore, assist in focusing on rational, analytical, and interdisciplinary research objectives that can aid in uncovering and critically analyzing "actual" examples of alternative and innovative learning [56]. The outputs are intended to educate decision-

making, encourage collaborative action, and promote thoughtful leadership to achieve transformative change toward sustainability and resilient urban futures [60,65].

## 4. Discussion

The current research provides the findings of a review of the literature concerning the role of cultural influence on urban sustainability. Based on the SLR results quantitative and qualitative content-based analyses were performed. As a planned approach to sustainable development in cities, urban sustainability transformations are usually nonlinear complex interactions and impact a broad range of diverse processes [36].

To understand the cultural heritage aspect of urban life and how this has a role in organizing urban issues, a number of scholars [6,19,20,33] have advocated the significance of cultural heritage and the utilization of cultural assets in the promotion of long-term urban development. At the city scale, the preservation of cultural heritage demonstrates the need to strike a balance between social, environmental, and economic factors [33]. In some cases, it is considered a tool for integrative and inclusive urban development [6], despite the risk globalization poses to historical, cultural, and social roots through economic homogenization [6]. A practical implementation and a promise are seen in "Urban Living Labs" [12,21,35,65] as transformative innovations aimed at hastening urban sustainability through multistakeholder governance, an "explicit form of intervention", by relying on "knowledge and learning". Academics [26] acknowledge that knowledge-sharing and co-production are critical components for establishing sustainable green governance, stimulating civic engagement and meaningful capacity building. It is in this context that the "Urban Living Lab" is a unique collaborative and participatory paradigm for urban regeneration [19–21,35] based on a structured user co-creation method in public–private–people partnerships [31].

Studies have also identified knowledge gaps when addressing heritage communities [44,46]. Authors [21,44] argue that for a stronger national community, development strategies are built on a synergistic participatory framework that will create a more participatory and aware community. Studies [21,22] point to a lack of understanding about the significance of interactions between urban subsystems, for attaining community sustainability on numerous scales, in multiple sectors, and across multiple disciplines [21,22].

The role of nature in urban planning for achieving sustainability has also been documented in the literature [11]; there is a concern for a model for ecosystem-based solutions to societal issues such as people's well-being and socio-economic advancement [47]. Nature-based solutions and urban green spaces provide a setting for leisure, social contact, social inclusion, and overall health and well-being. However, there are significant gaps in training and action, therefore the authors [50,51] seek to propose a framework to guide and assist in implementation. Sustainable urban behaviors also have the potential to increase both people's well-being and the psychological foundation of sustainable cultures, but they have remained understudied [54]. Similarly, sustainable urban governance lacks a thorough systematic understanding of the interaction between government and citizens for sustainable development. Cultural heritage governance is largely embedded within the larger national policy, at times influenced by transnational institutions, and therefore lacks a participatory management approach [45]. The failure of urban governance to address sustainability became more apparent during the current pandemic, when a lack of co-ordination between the different tiers of governance radically impacted city management and governance [34,38]. By actively participating in constructing the "global urban imagination" that supports global urban governance, urban academics may help ensure that this move is toward more progressive urban futures [37].

To achieve a transformative change toward sustainability, a meaningful and interdisciplinary educational program in the urban realm has evolved into a complex socio-cultural dialogue that advocates for higher education. Academics have also shown concern over the existing knowledge gap in practical frameworks to assess methods for urban sustainability [60] that remains fragmented and lacks interdisciplinary perspectives [61]. As a result,

ensuring that sustainability is covered across courses and in an interdisciplinary manner is critical to promoting inclusive sustainable education in higher education.

Thus, significant gaps have appeared in the review concerning major areas that are of concern for uniform sustainable urban growth. The desired outcomes of a just and environmentally sustainable future will be possible only when cities are fashioned to acknowledge a given issue and build needed capacities [4].

## 5. Conclusions and Future Direction

The goal of this paper was to investigate the impact of culture on urban sustainability, concerning the cultural heritage aspect of urban life and how this have a role in organizing urban issues. The article combined the concepts of culture with urban sustainability, as informed by the Sustainable Development Goals. With increasing urbanization, cultural heritage plays an important role in designing development strategies for contemporary cities, which are often confronted with socio-cultural concerns related to the loss of tangible and intangible cultural resources. The socio-cultural urban framework is inadequate unless urban areas acknowledge strong community ties. Incorporating tangible and intangible cultural resources, as well as cultural professionals and creative practices, has the potential to lead to a more sustainable urban future.

The quantitative study using the Dimension database and VOS viewer analysis revealed how current cultural heritage research is documented and its relevance increasingly gaining ground under the patronage of transnational institutions such as UNESCO and UCLG. While developed countries are major contributors, it is clear that some developing countries are also starting to contribute. African and Latin American countries have lagged behind despite their rate of urbanization. Owing to the limitation of access to online libraries like Scopus, the results could have been more profound otherwise. In addition, the cultural heritage studies are more concentrated on the tangible than intangible heritage and climate-related sustainability. The systematic literature review (SLR) identified 52 publications, and the study was classified into three categories: city uniqueness and cultural diversity, social connectivity and urban sustainability, and multidisciplinary and city knowledge.

When addressing the link between cultural heritage and urban sustainability, this study reveals the fragmentation and diversity of arguments, with a current focus on sustainability discourses and the challenges that come with them. Built heritage and the role of local community governance have demonstrated inequities in city administration. Knowledge gaps have also been found when addressing heritage communities and their understanding of urban ecosystem and management, as well as sustainable urban behaviors, all of which have been understudied.

Additionally, transitioning to a more interdisciplinary educational program in the urban realm is necessary to make the system inclusive by involving key stakeholders in the knowledge-generating process. Again, gaps exist in intercultural comparative learning in urban sustainability programs.

Therefore, it should come as no surprise that due to the vast extent of both culture and urban sustainability, it is impossible to do the subject justice in a single assessment. As previously noted, it necessitates extended discourses spanning interdisciplinary dialogue across sectors and among practitioners, academics, stakeholders, and others.

The difficulty underlying the implementation of cultural policies is yet another limitation for most countries due to institutional incapacities. As a result, the "one size fits all" strategy is not feasible for all nations. Many economies are already straining under the COVID-19 crisis, and they lack trained personnel and co-ordination across scales of governance and management. Moreover, training, when borrowed from the West, is usually a mis-fit for other political and socio-economic environments, and a burden to the borrowing economy. Quantitative analysis has revealed that most of the creative, cultural, and sustainable urban programs are concentrated in developed countries having access to both

funds and training, a lack of which is a likely hindrance to poor countries' socio-cultural and economic progress.

Finally, as the world prepares for an uncertain future in terms of social and economic constraints, climate, and migration, urban academics have a duty to resolve the issues confronting global urban sustainability and cultural disputes.

A more focused approach to research on culture and the urban sustainability issues can be undertaken by enquiring into:

1. How cities can sustain their cultural distinctiveness and heritage in the global urban context.
2. The current state of knowledge and practice related to culture and urban sustainability.
3. A cross-cultural investigation of how individuals see the urban sustainability problems in the post-COVID-19 era.

**Author Contributions:** Conceptualization, S.N. and S.S.; methodology and formal analysis, S.N. and S.S.; investigation, S.N. and S.S.; resources, S.N.; data curation, S.N. Writing—original draft preparation, S.N.; writing—review and editing, S.N. and S.S.; visualization, S.N. All authors have read and agreed to the published version of the manuscript.

**Funding:** This research received no external funding.

**Institutional Review Board Statement:** Not applicable.

**Informed Consent Statement:** Not applicable.

**Data Availability Statement:** Not applicable.

**Acknowledgments:** The authors will like to thank friends and colleagues for their encouragement through the writing and editing process. Also, six anonymous reviewers provided extremely helpful suggestions on how to improve the initial submission.

**Conflicts of Interest:** The authors declare no conflict of interest.

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
