# Peer review of "The Role of Cultural Heritage in Promoting Urban Sustainability: A Brief Review"

_land, doi:10.3390/land11091508_

Round 1

Reviewer 1 Report (Previous Reviewer 1)

Much better. Thank you for investing the extra time into this article. It is very good.

Author Response

Thanks for your encouragement.

Reviewer 2 Report (Previous Reviewer 2)

The paper has been improved, yet it still lacks a deeper thinking and overall analysis in the results. You identify several gaps ok, but what is the main conclusion? what is the main contribution of this paper on the field?

Author Response

  1. The objective of the article was to investigate, the impact of culture on urban sustainability, concerning the cultural heritage aspect of urban life and how they have a role in organizing urban issues.
  2. At the city scale, the preservation of cultural heritage demonstrates the need to strike a balance between social, environmental, and economic factors
  3. For a more integrative and inclusive urban development, this provides the tool to evade the influence of globalization on cultural heritage
  4. First of all, there are not many studies that have undertaken an inquiry into the emerging knowledge of urban sustainability as an academic discipline. Therefore, a multidisciplinary approach is likely to fill this gap.
  5. Moreover, urban academics have a duty to resolve the issue confronting global urban sustainability and cultural disputes.
  6. Therefore, the current article is a timely contribution to the research field encompassing urban sustainability and cultural issues.

Reviewer 3 Report (Previous Reviewer 3)

Thank you for the opportunity of reading and reviewing your interetsing manuscript. The paper addresses a topic which is under the scope of the journal and relevant for a wide public. The article is basically a review on the topic of cultural heritage as driver of urban sustainability. The methodology used includes both bibliometric analysis and content analysis. The bibliometric part is not that good and as a general rule for this type of investigation, it brings little new information. However, the content analysis is relevant and brings interesting new insights.

I would suggest to revise the paper by keeping in mind to clearly present the research gap identified and how you intend to fill it via the present article.

Moreover, the database used for investigation i.e. JSTOR is not very commonly used and this may be a serious limitation.

Finally, there is a relatively limited number of references used for this kind of article, I would expect to refer more sources given the nature of a review article.

In conclusion, I recommend revising the paper according to the said limitations.

Good luck!

Author Response

1. 

  1. Thanks for your valuable suggestions. The gaps identified have already been addressed in brief under discussion, otherwise it will unnecessarily make the article lengthy.
  2. It is best to consider the present as the author has access limitations to other databases such as Scopus and Web of Science as already mentioned earlier.
  3. In conclusion, it has already been suggested that scope for future inquiry exists in evaluating cultural distinctiveness and heritage in the global urban context.

Reviewer 4 Report (Previous Reviewer 5)

Dear authors,

I regret that, upon reading the latest revision, I continue to find the paper overly general, rife with unsupported observations, lacking any real data, conceptually weak, and rather wordy while not saying very much. I don't think it can contribute to the field without a complete rethinking of the premise and focusing in on a substantial idea that pushes well beyond generalities.

Author Response

  1. Appreciate your valuable suggestions for the first draft that helped in the revision. I have accordingly modified and added sufficient information for this review.
  2. I apologize for not being able to edit further, but I modified as much as I could do on my end.

Reviewer 5 Report (Previous Reviewer 6)

Dear Authors

thank you for making the corrections in accordance with my comments. 

Author Response

Appreciate your valuable initial suggestions, and accepting the revised version.

This manuscript is a resubmission of an earlier submission. The following is a list of the peer review reports and author responses from that submission.

Round 1

Reviewer 1 Report

The Influence of Cultural Heritage in Maintaining Urban Sustainability is what it claims, and that is a brief review. However, based on what is offered, the title and abstract either need to be revised, or the paper as a whole expanded upon. While non-Western and indigenous approaches and concepts to heritage conservation are mentioned in passing, very little of this is discussed in any depth. The focus of the paper admits that relies on "European context studies", while there is so much more on the Americas, Asia, and various parts of Africa. This is necessary to consider because the majority of the world's population lives in cities of these continents.

Missing from the analysis are definitions on what the authors are defining as "urban", in contrast to "rural" or even "suburban". While globalization is mentioned in passing, the counter interests and movements for localization are missing, and these have had profound impacts on theories pertaining to urban sustainability. There is also a lack of discussion about heritage tourism in relation to the "Influence of Cultural Heritage", which ignores how the usage of cultural heritage has become such a prominent part of many urban economies and their goals for becoming sustainable.

Reviewer 2 Report

The abstract sets a rather interesting frame. However, the paper doesn't come up to these expectations. Some questions I would ask are: how the relevant studies were chosen, how the different aspects of urban environment are interrelated and interconnected. Also, the main target of the paper is not clear. Post-covid era in cities is another issue that is mentioned but not connected with the other aspects. The main problems of the paper refer to the results and discussion sections where they mainly seem like a combination of many different elements rather than a robust argumentation on the main objective and aims of the paper. So, this part needs rewriting.

Reviewer 3 Report

Thank you for the opportunity of reading and reviewing your manuscript. It is a review paper addressing the contribution of culture to the sustainable development of cities, and the role of education in this regard. Being a review, there are relevant bibliographic sources referenced and grouped according to the main topics. I consider the paper necessary and the authors clearly indicate the topic (although it is a niche one). I strongly recommend to use some bibliometric instruments to illustrate the topics. Several analyses can be performed e.g. authors, journals, impact, key words etc. I suggest using vosviewer but this is not the only software that can be used. More in-depth analysis will increase the scientific value and more visualisation will increase readability. Good luck!

Reviewer 4 Report

The article presents the synthesis of some recent essays, related to very general topics; the criteria of selection and aggregation of the contents define a simple cataloguing, which does not give back interpretative keys that critically deepen the anthropological and philosophical challenges underlying the theme of cultural heritage. No authors of reference, schools of thought, axes of critical research emerge, but a simple list of anonymous bibliographical references without a direct reference to the territories. It is a mechanical association according to main-stream keywords, without a critical filter and without the effort to set the heritage reasoning on landscapes and territories: a reasoning on anthropology and on the philosophy of cultural heritage that disregards geography, languages, religions and philosophies seems a rather sterile exercise.
The very scholastic and descriptive synthesis proposed here can be a basic synthesis, which can orient the authors towards a scholarly research on the topic with a critical slant and a clear geographical definition.

Reviewer 5 Report

Dear authors,

I appreciate the attempt to draw links between urban cultural heritage and sustainability. However, I have several key concerns with the manuscript.  While urban cultural heritage is a stated focus, the text provides very little evidence of understanding or examples that promote new knowledge in the field. The methodology is overly simplistic, resulting in a failure to acknowledge several literature lineages that stretch back many decades.

The Results section lacks cohesion and clarity and is not enlightening. It belatedly introduces a myriad of disconnected themes and seemingly randomly selected references.There's no link to how the methodology drives the content of this section.

I regret to say that I'm unable to support this paper. Overall, the manuscript suffers from lack of focus and lack of follow through on the outcomes promised in the Abstract and Introduction. It is repetitive, much too generic, and reads like a college term paper. It attempts far too much, and delivers very little of substance that isn't already readily available in the literature. Literature reviews are valid if they create new clarity, make connections where substantial gaps have occurred, and generally advance theory. Unfortunately, this paper simply hasn't delivered on the promise of an effective literature review.

Reviewer 6 Report

Thank you for giving me the opportunity to review the article „ The Influence of Cultural Heritage in Maintaining Urban Sustainability: A Brief Review”. Below my remarks:

  • Authors have chosen an interesting and current topic on the Influence of cultural heritage in maintaining urban sustainability;
  • “The paper has presented an overview of the major thematic areas of enquiry in urban sustainability and culture. The study aims to determine how social and cultural factors play a critical role in urban sustainability and how multi-disciplinary education can help organize urban issues” - despite this reference to the aim of the article - I suggest in the abstract and in the introduction to formulate a clear aim of the article;
  • In the final part of the Introduction, please describe synthetically what the individual sections of the article contain;
  • "Data Collection and Selection Criteria" - Please justify why article search in Google Scholar database was chosen? This approach raises doubts. Why didn't the authors use databases such as Scopus or Web of Science? I suggest that the authors search additional databases and incorporate the results obtained into this article.
  • The article could be enriched with a quantitative analysis related to the material studied, presenting, for example, the most productive authors, countries, organizations, journals publishing within this topic. It would also be possible to add a map of keywords appearing in the analyzed articles.
  • in conclusion, authors should also describe the limitations of the study.
  • References should be properly prepared. E.g.
  1. Author 1, A.B.; Author 2, C.D. Title of the article. Abbreviated Journal Name Year, Volume, page range.
  2. Author 1, A.; Author 2, B. Title of the chapter. In Book Title, 2nd ed.; Editor 1, A., Editor 2, B., Eds.; Publisher: Publisher Location, Country, 2007; Volume 3, pp. 154–196.
  3. Author 1, A.; Author 2, B. Book Title, 3rd ed.; Publisher: Publisher Location, Country, 2008; pp. 154–196.